# Activation of the α7 Nicotinic Acetylcholine Receptor Prevents against Microglial-Induced Inflammation and Insulin Resistance in Hypothalamic Neuronal Cells

**DOI:** 10.3390/cells11142195

**Published:** 2022-07-14

**Authors:** Camila Libardi do Amaral, Ísis de Cássia Alves Martins, Alana Carolina Costa Veras, Fernando Moreira Simabuco, Michael Glenn Ross, Mina Desai, Leticia Martins Ignácio-Souza, Marciane Milanski, Adriana Souza Torsoni, Marcio Alberto Torsoni

**Affiliations:** 1Laboratory of Metabolic Disorders, School of Applied Sciences, University of Campinas, Limeira 13484-350, Brazil; camilalibardi09@gmail.com (C.L.d.A.); isismartins15@yahoo.com.br (Í.d.C.A.M.); alana_veras_@hotmail.com (A.C.C.V.); llmisouza@unicamp.br (L.M.I.-S.); marcia31@unicamp.br (M.M.); atorsoni@unicamp.br (A.S.T.); 2Multidisciplinary Laboratory of Food and Health, School of Applied Sciences, University of Campinas, Limeira 13484-350, Brazil; ferms@unicamp.br; 3The Lundquist Institute, David Geffen School of Medicine, Harbor-UCLA Medical Center, University of California, Los Angeles, CA 90095, USA; mikeross@ucla.edu (M.G.R.); mdesai@lundquist.org (M.D.); 4Obesity and Comorbidities Research Center, University of Campinas, Campinas 13083-864, Brazil

**Keywords:** α7nAChR, inflammation, insulin resistance, hypothalamus, neurodegeneration

## Abstract

Neuronal hypothalamic insulin resistance is implicated in energy balance dysregulation and contributes to the pathogenesis of several neurodegenerative diseases. Its development has been intimately associated with a neuroinflammatory process mainly orchestrated by activated microglial cells. In this regard, our study aimed to investigate a target that is highly expressed in the hypothalamus and involved in the regulation of the inflammatory process, but still poorly investigated within the context of neuronal insulin resistance: the α7 nicotinic acetylcholine receptor (α7nAchR). Herein, we show that mHypoA-2/29 neurons exposed to pro-inflammatory microglial conditioned medium (MCM) showed higher expression of the pro-inflammatory cytokines IL-6, IL-1β, and TNF-α, in addition to developing insulin resistance. Activation of α7nAchR with the selective agonist PNU-282987 prevented microglial-induced inflammation by inhibiting NF-κB nuclear translocation and increasing IL-10 and tristetraprolin (TTP) gene expression. The anti-inflammatory role of α7nAchR was also accompanied by an improvement in insulin sensitivity and lower activation of neurodegeneration-related markers, such as GSK3 and tau. In conclusion, we show that activation of α7nAchR anti-inflammatory signaling in hypothalamic neurons exerts neuroprotective effects and prevents the development of insulin resistance induced by pro-inflammatory mediators secreted by microglial cells.

## 1. Introduction

Neuroinflammation is an inflammatory process that occurs in response to potentially harmful stimuli to the central nervous system [1]. One of the main events related to the onset of neuroinflammation is the activation of microglial cells, the resident macrophages of the central nervous system [2]. Once activated, microglial cells release pro-inflammatory mediators such as interleukin 6 (IL-6), interleukin 1β (IL-1β), tumor necrosis factor α (TNF-α), and fractalkine (CX3CL1), which are essential for protecting the central nervous system and maintaining its homeostasis. However, the uncontrolled and prolonged liberation of these inflammatory mediators can exert several deleterious effects on adjacent neuronal cells, including the development of insulin resistance [3].

Central insulin signaling is particularly important in the hypothalamus, where it plays a crucial role in the regulation of glycemic homeostasis and energy metabolism [4]. It is widely recognized that obesity is associated with the development of both central and peripheral insulin resistance. On the other hand, the interruption of hypothalamic insulin signaling also compromises the central control over peripheral metabolism and contributes to the development of several metabolic diseases, including obesity and type 2 diabetes mellitus [5,6]. Chronic hyperglycemia induced by insulin resistance also exerts deleterious effects on neurological functions and is related to the development of cognitive deficit and dementia [7]. Impaired insulin signaling and consequent inhibition of PI3K/Akt pathway activity lead to the activation of GSK-3 kinase and the phosphorylation of tau, a microtubule-associated protein found primarily in the axons of neurons [8,9]. Hyperphosphorylation of tau causes microtubule destabilization and the formation of neurofibrillary tangles, one of the main pathological features of Alzheimer’s disease [10].

Considering that the development of hypothalamic insulin resistance is closely linked to neuroinflammation, several targets involved in the control of the inflammatory response have been studied as a potential therapeutic strategy in the prevention of insulin resistance and associated metabolic and neurological disorders. In this regard, our study is focused on the investigation of a target involved in the regulation of the inflammatory process [11] and widely expressed in the central nervous system, but still poorly studied in the context of neuronal insulin resistance: the α7 nicotinic acetylcholine receptor (α7nAchR). α7nAchR is a ligand-gated ion channel that can be found not only in neurons but also in non-neuronal cells, particularly lymphocytes and macrophages [11,12]. Activation of α7nAchR by nicotine, acetylcholine, and other agonists inhibits the transcription of pro-inflammatory cytokines by activating the JAK2/STAT3 signaling pathway [13], playing an important role in immunity, inflammation, and neuroprotection.

In the past few years, our group has shown that α7nAchR activation in mice hypothalamus plays an important role in regulating the expression of anorexigenic neuropeptides, resulting in the reduction of food intake [14]. However, hypothalamic expression of α7nAchR can be impaired under pro-inflammatory conditions. Recently, we reported that Swiss mice fed with a high-fat diet (HFD) for only three days show a marked reduction in the expression of α7nAchR in the hypothalamus, accompanied by an increase in the expression of pro-inflammatory markers, such as NF-κB, and a reduction in anti-inflammatory markers, such as interleukin 10 (IL-10) [15].

Although our studies show that α7nAchR expression in the hypothalamus is negatively and early modulated under pro-inflammatory conditions, its role in inflammation and insulin resistance specifically in hypothalamic neurons remains poorly investigated. Given this, we hypothesize that activation of α7nAchR in hypothalamic neurons can prevent the development of inflammation-induced insulin resistance, as well as prevent cellular events related to neurodegeneration.

## 2. Materials and Methods

### 2.1. Animal Experiments

Male C57BL/6J mice (*Mus musculus*) (7 weeks old, 30–40 g body weight) were randomly distributed into 2 groups of 5 individuals: one was fed a standard chow diet (SC) (Nuvilab^®^ CR-1, Nuvital, PR, Curitiba, Brazil) and the other was fed a high-fat diet (HFD; 60%) for 3 days. The HFD was prepared in our laboratory according to the AIN-93G standard modified for high-fat (60%) content (Table 1). The mice were kept under controlled temperature (22–24 °C) and a 12 h light/dark cycle, with access to water and food ad libitum. After receiving each diet for 3 days, the mice were anesthetized (100 mg/kg ketamine of body weight and 100 mg/kg xylazine of body weight, i.p.) and euthanized for the extraction of the hypothalamus. The animals were provided by the Animal Breeding Center at the University of Campinas (UNICAMP) and the experimental procedures were approved by the Ethics Committee on Animal Use (ECAU) of UNICAMP (ID protocol 5432-1) and performed according to the guidelines of the Brazilian Society of Laboratory Animal Science (SBCAL).

### 2.2. Cell Culture

BV-2 murine microglial cells (RRID:CVCL_0182) and mHypoA-2/29 (RRID:CVCL_D394) murine hypothalamic neurons were cultured in Dulbecco’s Modified Eagle’s Medium (DMEM) High Glucose (Sigma-Aldrich, St. Louis, MO, USA) supplemented with 10% FBS (Sigma-Aldrich, St. Louis, MO, USA) and 1% penicillin-streptomycin solution (100 U/mL; 100 µg/mL) (Sigma-Aldrich, St. Louis, MO, USA). mHypoA-POMC/GFP (RRID:CVCL_EP69) murine hypothalamic neurons were cultured in Dulbecco’s Modified Eagle’s Medium (DMEM) Low Glucose supplemented with 5% FBS and 1% penicillin-streptomycin solution (100 U/mL; 100 µg/mL). mHypoA-2/29 and mHypoA-POMC/GFP cell lines were immortalized from C57/BL6 mice [16] and kindly provided by Dr. Denise Belsham, University of Toronto. All cell lines were maintained in standard culture conditions (37 °C, 5% CO_2_, and 95% relative humidity). To assess Chrna7 gene expression by RT-qPCR, mHypoA-POMC/GFP cells were treated with 100 ng/mL lipopolysaccharide (LPS derived from *E. coli* 0111:B4; Sigma-Aldrich, St. Louis, MO, USA) for 3 h.

### 2.3. Microglial Conditioned Medium (MCM) Preparation

The conditioned medium preparation protocol was adapted from Jeong et al. [17] and Duffy et al. [18]. To induce microglial inflammation, BV-2 cells were treated with 100 ng/mL LPS for 3 h or left unstimulated (control). LPS concentration and incubation time were determined by time-course experiments (data not shown). The cells were then washed twice with PBS to remove any residual LPS and maintained in fresh culture medium for 16 h. After that, the culture medium was collected, centrifuged at 2000 rpm for 10 min, and used as microglial conditioned medium (MCM) for mHypoA-2/29 cells.

### 2.4. ELISA

ELISA kits were used to measure the concentration levels of the proinflammatory cytokines IL-6 (7.8–500 pg/mL; #KMC0062; Invitrogen, Carlsbad, CA, USA; dilution: 1:40), IL-1β (15.6–1000 pg/mL; #KMC0011; Invitrogen, Carlsbad, CA, USA; dilution: 1:2) and TNF-α (31.3–2000 pg/mL; #DY410; R&D Systems, McKinley Place Northeast, MN, USA; dilution: 1:40) in the microglial conditioned medium. All tests were performed following each manufacturer’s instructions.

### 2.5. Treatment of mHypoA-2/29 Neuronal Cells with MCM

mHypoA-2/29 neuronal cells were incubated for 6 h with the conditioned medium obtained from microglial cells previously stimulated with LPS (MCM-LPS) or from unstimulated microglial cells (MCM-C). For some experiments, TLR4 specific inhibitor TAK-242 (10 µM; 1 h) was added before the conditioned medium. To maintain α7nAchR activation throughout the treatment period, a stimulus with the selective agonist PNU-282987 was carried out 15 min before and 3 h after the addition of the conditioned medium. Some groups were stimulated with JAK2 inhibitor AG-490 (100 µM 30 min; Cayman Chemical, Ann Arbor, MI, USA), STAT3 specific inhibitor Stattic (30 µM 30 min; Cayman Chemical, Ann Arbor, MI, USA), or α7aAchR specific antagonist α-bungarotoxin (100 µM 1 h; Cayman Chemical, Ann Arbor, MI, USA) before the addition of PNU-282987. At the end of the conditioned medium treatment, some groups were stimulated with 100 nM insulin for 10 min.

### 2.6. Transfections

mHypoA-2/29 cells were seeded into 60 mm culture dishes and cultured until reaching 70% confluency. To induce plasmid-driven overexpression of α7nAChR, the cells were transfected with pCMV6-entry plasmid carrying an α7nAChR Open Reading Frame (OriGene, Rockville, MD, USA; #MR224522) using TurboFectin™ 8.0 reagent (OriGene, Rockville, MD, USA). For α7nAch knockdown, mHypoA-2/29 cells were transfected with a pool of three Chrna7-specific siRNA sequences (Santa Cruz Biotechnology, Santa Cruz, CA, USA, #sc-42533) using Lipofectamine RNAiMAX (Thermo Scientific, Waltham, MA, USA). Transfections were performed according to each manufacturer’s instructions. Forty-eight hours after transfection, the cells were incubated with the microglial conditioned medium (MCM), as described above.

### 2.7. CRISPR-Cas9-Mediated Knockout of Chrna7

A single guide RNA (sgRNA) targeting the Chrna7 gene, which encodes the α7 subunit of the nicotinic acetylcholine receptor (α7nAchR), was designed using CRISPOR [19]. sgRNA oligos (F: 5′-CACCGGTGAGCGGCTGCGAGTCGT-3′; R: 5′-AAACACGACTCGCAGCCGCTCACC-3′) were cloned into the Cas9 expression vector pSpCas9(BB)-2A-Puro (PX459) V2.0 (Addgene, Watertown, MA, USA; #62988) and transfected into mHypoA-2/29 cells using Lipofectamine and Plus Reagent (Thermo Scientific, Waltham, MA, USA), according to Dr. Zhang’s protocol [20]. Transfected cells were selected with puromycin (1 µg/mL) 72 h after transfection. Genomic DNA was extracted and CRISPR-Cas9 cleavage efficiency was assessed by the T7 endonuclease I assay. Single cells were isolated by serial dilutions in 96-well plates and the resulting monoclonal cell lines were analyzed by Sanger sequencing.

### 2.8. Western Blotting

Whole-cell protein lysates were obtained using 1x Cell Lysis Buffer (Cell Signaling Technology, Danvers, MA, USA) containing 1 mM PMSF (Sigma-Aldrich, St. Louis, MO, USA), 1% phosphatase inhibitor cocktail 2 (Sigma-Aldrich, St. Louis, MO, USA), and 1% protease inhibitor (Sigma-Aldrich, St. Louis, MO, USA). Hypothalamic protein lysates were obtained by homogenizing the tissues in freshly prepared ice-cold buffer [1% (*v*/*v*) Triton X-100, 0.1 M Tris, pH 7.4, 0.1 M sodium pyrophosphate, 0.1 M sodium fluoride, 0.01 M EDTA, 0.01 M sodium vanadate, and 0.002 M PMSF and 0.01 M aprotinin]. Lysates were centrifuged at 12,000× *g* for 10 min at 4 °C and the supernatant was collected. Separation of nuclear and cytoplasmatic protein extracts was performed using Abcam’s Nuclear Extraction Kit (Abcam, Cambridge, UK; ab113474). For all samples, protein concentration was measured using the Bradford dye-binding method. Protein lysates were boiled in Laemmli buffer for 5 min, resolved on 10% SDS-PAGE gels, and then blotted onto 0.22 µM PVDF membranes (Bio-Rad, Mississauga, ON, Canada). Membranes were blocked at room temperature for 2 h in 5% nonfat dry milk containing 0.2% Tween-20 and incubated overnight at 4 °C with the following primary antibodies: p-Akt S473 (Cell Signaling Technology, Danvers, MA, USA, #4060S; 1:1000); p-STAT3 Y705 (Cell Signaling Technology, Danvers, MA, USA, #9145S; 1:1000); p-NF-κB p65 S536 (Cell Signaling Technology, Danvers, MA, USA, #3033S; 1:1000); p-GSK3α S21 (Santa Cruz Biotechnology, Santa Cruz, CA, USA, #sc-81497; 1:1000); p-tau Ser726 (Santa Cruz Biotechnology, Santa Cruz, CA, USA, #sc-16945; 1:1000); GAPDH (Santa Cruz Biotechnology, Santa Cruz, CA, USA, #sc-25778; 1:1000). Following 3 × 5 min washes with TBS-T (Tris-buffered saline, 0.1% Tween 20), membranes were incubated with HRP-conjugated secondary antibodies (KPL, Gaithersburg, MD, USA) for 2 h at room temperature. Blots were exposed to the SuperSignal West Pico Chemiluminescent Substrate (Thermo Scientific, Waltham, MA, USA) and immunoreactive bands were visualized using the GeneGnome XRQ chemiluminescence imaging system (Syngene, Frederick, MD, USA). Band intensity was quantified using Image J software and normalized to the endogenous control (GAPDH).

### 2.9. Quantitative PCR Analysis

Total RNA was isolated from cells using Trizol Reagent (Thermo Scientific, Waltham, MA, USA) according to the manufacturer’s instructions. RNA concentration and purity were measured using the NanoDrop 2000 spectrophotometer (Thermo Scientific, Waltham, MA, USA). Reverse transcription was performed with 3 µg of total RNA using the High Capacity cDNA Reverse Transcription kit (Applied Biosystems, Foster City, CA, USA). Real-time quantitative PCR assays were performed using the ABI Prism 7500 Fast Sequence Detection System. Zpf36 gene expression levels were determined using the SYBR Green™ detection system with the following forward and reverse qPCR primers: F: AAATTCAGTGTTTGGGTGGA; R: TGTAACCCCAGAACTTGGAA. All other reactions were performed using TaqMan™ probes (Applied Biosystems, Waltham, MA, USA): Chrna7: Mm01312230_m1; Il6: Mm00446190_m1; Il1b: Mm00434228_m1; Tnf: Mm00443258_m1; Il10: Mm01288386_m1; Cx3Cl1: Mm00436454_m1; Gapdh: #4352339E. Relative expression levels of target genes were calculated by the ΔΔCt method, using Gapdh as the housekeeping reference gene.

### 2.10. Statistical Analysis

Results were expressed as mean ± SEM and are representative of at least three independent experiments. Student’s unpaired *t*-tests were used to compare the differences between two groups. One-way analysis of variance (ANOVA) followed by multiple comparisons between groups using Tukey’s post-hoc test was used when differences between more than two groups were analyzed. The criterion determined for statistical significance was * *p* < 0.05, ** *p* < 0.01, *** *p* < 0.001, and **** *p* < 0.0001. Statistical analyses were performed using GraphPad Prism^®^ software (Version 8.0 GraphPad Software, Inc.; http://www.graphpad.com/scientificsoftware/prism/, accessed on 26 May 2022).

## 3. Results

### 3.1. Short-Term High-Fat Diet and LPS Reduce α7nAchR Expression in Mice Hypothalamus and Hypothalamic Neurons

Considering that HFD-induced inflammation reduces α7nAchR expression in the hypothalamus of Swiss mice, we first evaluated the impact of short-term HFD exposure on α7nAChR expression in the hypothalamus of C57BL/6J mice. As expected, consumption of HFD for just 3 days significantly reduced the mRNA levels of the gene that encodes the α7 receptor (Chrna7) in the hypothalamus by 75% (±16.6) when compared to the SC group (Figure 1A). Next, we investigated the effect of inflammation on Chrna7 expression specifically in neuronal cells using a culture of immortalized hypothalamic neurons and observed that the mRNA expression of Chrna7 was reduced by 60% (±9.5) in LPS-stimulated mHypoA-POMC/GFP neurons when compared to control cells (Figure 1B).

### 3.2. Activation of BV-2 Cells with LPS Induces the Formation of Pro-Inflammatory Microglial Conditioned Medium (MCM)

To better understand the role of α7nAChR in hypothalamic neuronal inflammation and reproduce the physiological interactions that occur between neurons and microglia cells in the context of neuroinflammation, we used the conditioned medium of LPS-activated BV-2 microglial cells to induce inflammation in mHypoA-2/29 neurons. Activation of BV-2 cells with LPS (100 ng/µL) for 3 h significantly enhanced the mRNA levels of the pro-inflammatory cytokines IL-6, IL-1β, and TNF-α (Figure 2A–C). After LPS treatment, the cells were washed and incubated with fresh culture medium for 16 h and the presence of pro-inflammatory cytokines in the supernatant was analyzed. The conditioned medium obtained from BV-2 cells previously treated with LPS (MCM-LPS) showed significantly increased levels of IL-6, IL-1β, and TNF-α when compared to conditioned medium obtained from cells maintained in culture medium only (MCM-C) (Figure 2D–F), confirming that the pre-exposure to LPS was sufficient to condition microglia to a more pro-inflammatory profile.

### 3.3. Microglial Conditioned Medium Induces Inflammation in mHypoA-2/29 Neurons

Next, mHypoA-2/29 neurons were incubated with MCM-LPS or MCM-C for 6 h and the effect of the treatment with the microglial conditioned medium on the expression of inflammatory mediators was evaluated. MCM-LPS treatment significantly increased the expression of IL-6, IL-1β, TNF-α, and the pro-inflammatory chemokine CX3CL1 in mHypoA-2/29 cells (Figure 3A–D). No changes were observed in IL-10 expression levels (Figure 3E). Importantly, the increased expression of inflammatory markers was concomitant with a 55% reduction in Chrna7 expression (Figure 3F).

To ensure that inflammation was not caused by possible LPS residues remaining in the conditioned medium, mHypoA-2/29 cells were pretreated with TLR4 inhibitor TAK-242 (10 µM; 1 h) and then incubated with LPS (100 ng /mL; 3 h) or with MCM-LPS (6 h). As expected, TAK-242 treatment significantly inhibited the expression of all cytokines analyzed in LPS-treated cells, but not in the MCM-LPS-treated group (Figure 3G–I). Therefore, these results confirm that MCM-LPS was effective in inducing neuronal inflammation by exposure to cytokines, chemokines, and other inflammatory mediators secreted by activated microglia.

### 3.4. α7nAchR Activation Protects Neurons against Microglial-Induced Inflammation by Inhibiting NF-κB Nuclear Translocation and Activating TTP Gene Expression

The effect of α7nAchR activation on neuronal inflammation was investigated using its selective agonist PNU-282987. Treatment with PNU attenuated MCM-LPS-induced increased gene expression of IL-6, IL-1β, TNF-α, and CX3CL1 (Figure 4A–D). This was accompanied by an increase in the expression of the anti-inflammatory cytokine IL-10 (Figure 4E) and the RNA-binding protein tristetraprolin (TTP; encoded by Zpf36 gene) (Figure 4F) and by a reduction in the nuclear translocation of NF-κB (Figure 4G).

To confirm that these effects were mediated by α7nAchR, we developed a mHypoA-2/29 knockout cell line via CRISPR/Cas9. Knockout confirmation was determined by analyzing STAT3 phosphorylation after treatment with a α7nAchR selective agonist. mHypoA-2/29 knockout neurons proved to be unresponsive to PNU, confirming the loss of α7nAchR functionality in these cells (Figure 5A). Importantly, the expression of the proinflammatory cytokines IL-6, IL-1β, and TNF-α (Figure 5B–D) was not reduced following PNU treatment in knockout neurons exposed to MCM-LPS. Furthermore, PNU also failed to increase IL-10 and TTP expression in knockout neurons (Figure 5E–F). Taken together, these results indicate that α7nAchR activation prevents microglia-induced neuronal inflammation by mechanisms that involve a reduction in NF-κB nuclear translocation and an increase in TTP gene expression.

### 3.5. Activation of α7nAchR Prevents Insulin Resistance and GSK3/tau Activation in Neurons

Considering that inflammation is associated with the development of insulin resistance in neurons, we evaluated the effect of MCM-induced inflammation on insulin signaling in mHypoA-2/29 cells. Akt phosphorylation levels in response to insulin were significantly reduced in the MCM-LPS group when compared to the MCM-C group, indicating a state of reduced insulin sensitivity (Figure 6A). Activation of α7nAchR with PNU completely prevented insulin resistance induced by MCM-LPS (Figure 6A).

To ensure that this improvement in insulin sensitivity was due to the activation of α7nAchR, we investigated the effect of α7nAchR expression on neuronal insulin resistance. While plasmid-driven overexpression of α7nAchR sustained PNU-induced improvement in insulin sensibility, knockdown of α7nAchR gene expression by siRNA completely abrogated its effects, providing evidence that α7nAchR improves insulin sensitivity in neuronal cells (Figure 6B). To investigate the mechanism by which α7nAchR activation prevents insulin resistance, we also evaluated the ability of PNU to prevent insulin resistance after pharmacological inhibition of α7nAchR and its downstream effectors JAK2 and STAT3. We observed that α-bungarotoxin (α7nAchR-selective antagonist), AG-490 (JAK2-specific inhibitor), and Stattic (STAT3-specific inhibitor) completely blocked the effect of PNU of preventing insulin resistance in mHypoA-2/29 neurons, indicating that the improvement of insulin signaling promoted by α7nAchR involves JAK2 and STAT3 signaling (Figure 6C).

Due to the importance of hypothalamic insulin signaling in the regulation of neuromodulatory and neuroprotective processes, we investigated the effect of α7nAchR activation on the phosphorylation of GSK-3 and tau. Phosphorylation of GSK-3 in the MCM-LPS group was found to be significantly reduced when compared to the MCM-C group, regardless of the presence of insulin (Figure 6D). This effect was blocked by the activation of α7nAchR with PNU (Figure 6D). Accordingly, the levels of tau phosphorylation, which were significantly increased in the MCM-LPS group, were restored to values close to those in the control group after PNU treatment (Figure 6E). Taken together, the results suggest a protective effect of α7nAchR against neurodegeneration in hypothalamic neurons.

## 4. Discussion

Insulin resistance in hypothalamic neurons is associated with several changes in energy metabolism, in addition to being an important factor in the pathogenesis of neurodegenerative diseases [4,21,22]. In our study, a model of neuroinflammation and insulin resistance was reproduced in vitro by culturing mHypoA-2/29 neurons with the pro-inflammatory conditioned medium of activated BV-2 microglia cells. In addition to simulating the physiological interactions that occur between neurons and microglia cells during the inflammatory process, the conditioned medium system also makes it possible to individually examine cellular changes in different lineages [23].

Here, we show that α7nAchR expression is reduced in immortalized hypothalamic neurons directly exposed to LPS or treated with pro-inflammatory microglial conditioned medium. A7nAchR expression was also shown to be markedly reduced in the total hypothalamic extract of C57/B6 mice fed with HFD for only 3 days. Recently, our group found similar results using Swiss mice, in which the expression and protein content of α7nAchR in the hypothalamus and other tissues were impaired by intraperitoneal injection of LPS or consumption of short-term HFD [15,24].

Considering that α7nAchR is a key regulator of cytokine release, we used its selective agonist PNU-282987 to try to overcome a possible worsening of inflammation caused by the loss of α7nAchR expression. Here, we showed that the increased expression of IL-6, IL-1β, TNF-α, and Cx3cl1 in neuronal cells exposed to pro-inflammatory microglial conditioned medium was completely abolished by the activation of α7nAchR with PNU. In agreement with these results, we also observed that PNU reduced the levels of nuclear phosphorylation of NF-κB, an important transcriptional promoter of these genes [25,26]. Other studies have also observed that α7nAchR activation negatively regulates the nuclear translocation of NF-κB and attenuates the production of inflammatory cytokines in other CNS cells, including microglia [27] and astrocytes [28].

Among the mechanisms involved in the regulation of the inflammatory response by α7nAchR is the activation of the intracellular signaling pathway of JAK2/STAT3. The anti-inflammatory function of STAT3 is closely related to the activity of the cytokine IL-10 [29]. STAT3 has a specific binding site in the promoter region of the gene that encodes IL-10 and can control its transcription [30,31,32]. Therefore, it is possible that the increase in IL-10 expression observed in our study is due to the transcriptional activity of p-STAT3.

IL-10 also acts as an important regulator of the anti-inflammatory activity of STAT3. Intracellular IL-10 signaling leads to the activation of JAK1, which phosphorylates STAT3. Upon being phosphorylated, STAT3 dimerizes and translocates to the cell nucleus, where it promotes the transcription of target genes that act by repressing the expression of pro-inflammatory genes [29]. Among the genes activated by STAT3 is Zfp36, which encodes tristetraprolin (TTP), an RNA-binding protein (RBP) that participates in the post-transcriptional regulation of gene expression [33,34]. TTP recognizes and binds to adenine and uracil-rich regions (AREs) in the 3′ untranslatable region (UTR) of the messenger RNA of target genes, promoting its degradation [33]. In this sense, our results indicate that the anti-inflammatory action exerted by PNU can also be attributed to the degradation of IL-6, IL-1β, and TNF-α mRNA via STAT3/TTP activation.

To prove the involvement of IL-10 and TTP in the anti-inflammatory response induced by the activation of α7nAchR signaling in neurons, we developed a mHypoA-2/29 knockout cell line for the Chrna7 gene, which encodes the α7 receptor. Due to the lack of specificity of commercially available antibodies for the α7 receptor, knockout confirmation was based on p-STAT3 levels after PNU stimulation [35,36]. As expected, we observed that the knockout of α7nAchR abolished the effect of PNU to prevent the increase in the expression of Il-6, Il-1β, and TNA-α induced by pro-inflammatory microglial conditioned medium in neurons. Furthermore, the expression levels of IL-10 and TTP remained unchanged after stimulation with PNU in the knockout group, proving the participation of IL-10/TTP signaling in the anti-inflammatory response mediated by the α7 receptor. Other studies have shown that nicotine—a non-selective agonist of nicotinic acetylcholine receptors—increases TTP expression and reduces TNF-α production in LPS-stimulated macrophages [37,38]. These effects are abolished with STAT3 knockdown via siRNA [38]. Considering that most studies on the anti-inflammatory function of the α7 receptor focus on macrophages and microglia cells, our work contributes by demonstrating for the first time the important role of α7nAchR in the regulation of TTP expression also in neuronal cells.

By using pharmacological and genetic approaches, our study reported that α7nAchR anti-inflammatory signaling also contributed to preventing the development of insulin resistance in neurons treated with microglial conditioned medium. We show that α7nAchR overexpression and activation with selective agonist activation prevent the development of inflammation-induced insulin resistance in hypothalamic neurons. This effect is lost due to the inhibition of α7nAchR gene expression and the pharmacological inhibition of the receptor and proteins downstream of its signaling pathway, which proves that its protective effect is indeed exerted by the activation of the JAK2/STAT3 pathway.

Due to the importance of hypothalamic insulin signaling in the regulation of neuromodulatory and neuroprotective processes, the development of insulin resistance in the hypothalamus is often associated with neurodegenerative events [39]. By improving neuronal insulin signaling, α7nAchR also contributed to reducing the activation of important neurodegeneration markers such as GSK3 and tau, which suggests a protective effect of α7nAchR against neurodegeneration of hypothalamic neurons. In agreement with our results, it was recently shown that α7nAchR activation with selective agonist increases Akt phosphorylation and attenuates GSK3β activation in the cortex and hippocampus of mice injected with streptozotocin [40]. This improvement in insulin signaling was accompanied by reduced expression of inflammatory markers and improved memory and cognitive function [40]. Additionally, cognitive decline in 20-week-old obese and insulin-resistant rats is associated with reduced α7nAchR protein content in the frontal cortex and hippocampus [41,42]. Finally, as it represents an early stage of the hypothalamic inflammatory process [15], we believe that reduced α7nAchR receptor expression may be a critical factor in making neuronal cells more susceptible to the development of dementia and other neurodegenerative diseases, such as Alzheimer’s.

## 5. Conclusions

Our study shows that activation of the α7 nicotinic acetylcholine receptor in hypothalamic neuron culture prevented inflammation induced by microglia cells by mechanisms involving a reduction in NF-κB nuclear translocation and an increase in TTP gene expression. The anti-inflammatory effect of the α7 receptor was accompanied by an improvement in neuronal insulin sensitivity and lower activation of proteins related to neurodegeneration.

## Figures and Tables

**Figure 1 cells-11-02195-f001:**
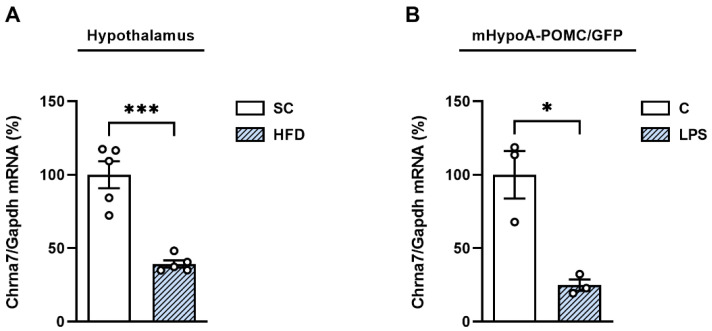
Short-term high-fat diet and LPS reduce Chrna7 expression in mice hypothalamus and hypothalamic neurons. (**A**) RT-qPCR analysis of α7nAchR expression levels in the hypothalamus of C57BL/6J mice fed with HFD (60%) or standard chow (SC) for 3 days and (**B**) in mHypoA-POMC/GFP immortalized hypothalamic neurons treated with LPS (100 ng/mL) for 3 h. * *p* < 0.05, *** *p* < 0.001, n = 5-3.

**Figure 2 cells-11-02195-f002:**
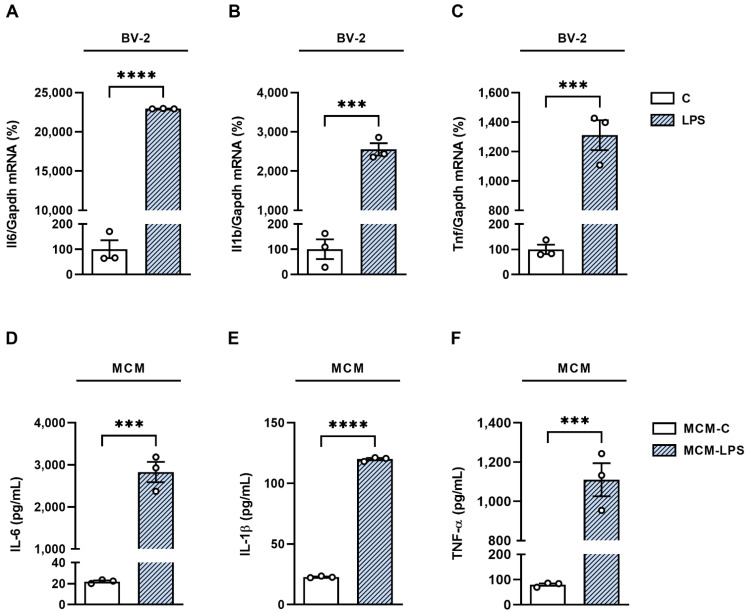
Activation of BV-2 cells with LPS induces the formation of pro-inflammatory microglial conditioned medium (MCM). BV-2 cells were treated with LPS (100 ng/mL) for 3 h and the expression of mRNA of genes encoding the pro-inflammatory cytokines (**A**) IL-6, (**B**) IL-1β, and (**C**) TNF-α was evaluated by RTq-PCR. After LPS treatment, the cells were washed and kept in fresh culture medium for 16 h. The supernatant obtained from BV-2 cells previously treated with LPS (MCM-LPS) or maintained only in culture medium (MCM-C) was collected and the presence of the pro-inflammatory cytokines (**D**) IL-6, (**E**) 1L-1β, and (**F**) TNF-α was quantified by the ELISA method. *** *p* < 0.001, **** *p* < 0.0001, n = 3.

**Figure 3 cells-11-02195-f003:**
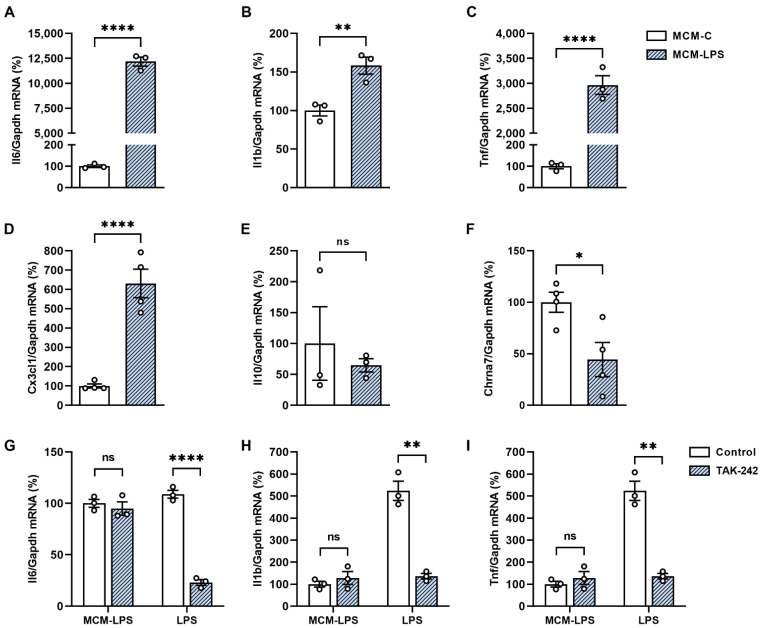
Microglial conditioned medium induces inflammation in neuronal cells. (**A**–**F**) mHypoA-2/29 immortalized neuronal cells were incubated for 6 h with microglial conditioned medium obtained from control BV-2 cells (MCM-C) or cells previously treated with LPS (MCM-LPS) and mRNA expression of the genes encoding (**A**) IL-6, (**B**) IL-1β, (**C**) TNF-α, (**D**) CX3CL1, (**E**) IL-10, and (**F**) Chrna7 were evaluated by RTq-PCR. (**G**–**I**) mHypoA-2/29 cells were treated with the TLR4 inhibitor TAK-242 (10 µM) 1 h before MCM-LPS or LPS treatment and the expression of (**G**) IL-6, (**H**) IL-1β, (**I**) TNF-α was evaluated by RTq-PCR. * *p* < 0.05, ** *p* < 0.01, **** *p* < 0.0001, n = 3-–4.

**Figure 4 cells-11-02195-f004:**
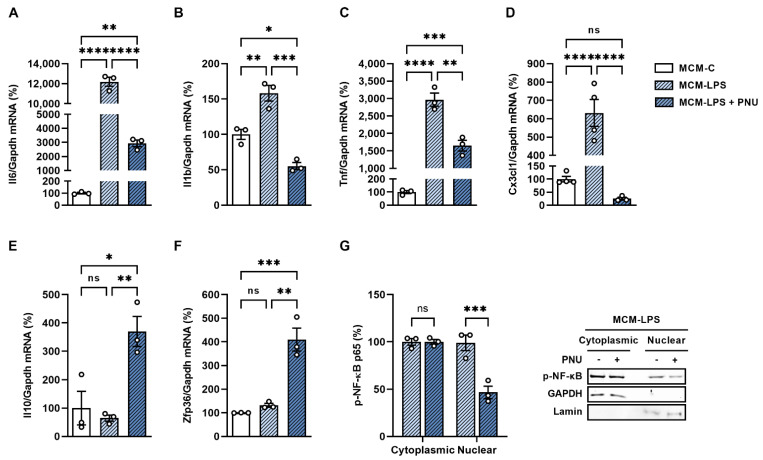
Activation of α7nAchR prevents inflammation induced by microglial conditioned medium and increases the expression of Il-10 and TTP in neurons. Immortalized mHypoA-2/29 neuronal cells were incubated for 6 h with MCM-C or MCM-LPS. A stimulus with the selective α7nAchR agonist PNU-282987 (1 µM) was administered 15 min before and 3 h after the start of conditioned medium treatment. After treatment, (**A**) IL-6, (**B**) IL-1β, (**C**) TNF-α, (**D**) CX3CL1, (**E**) IL-10, and (**F**) Zfp36 mRNA expression was evaluated by RTq-PCR and (**G**) NF-κB phosphorylation was evaluated by western blot. * *p* < 0.05, ** *p* < 0.01, *** *p* < 0.001, **** *p* < 0.0001, n = 3–4.

**Figure 5 cells-11-02195-f005:**
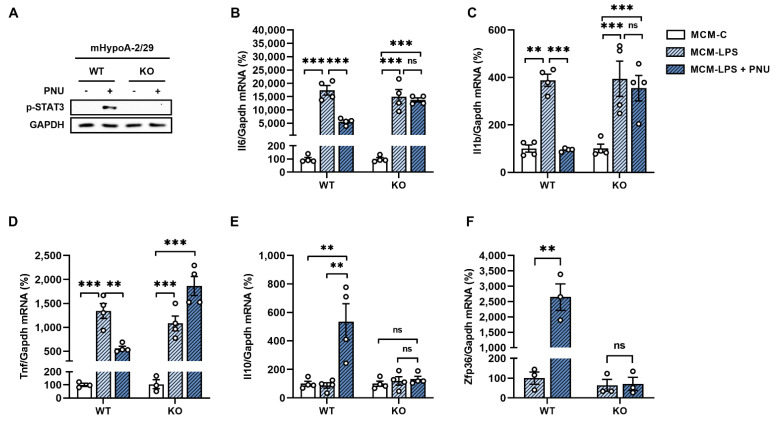
α7nAchR knockout inhibits the effect of PNU-282987 to increase IL-10 and TTP expression in neurons. An α7nAchR knockout mHypoA-2/29 cell line was established via CRISPR/Cas9. (**A**) The functionality of the α7nAchR receptor in WT and KO cells was evaluated by Western blot based on STAT3 phosphorylation levels after treatment with α7nAchR selective agonist PNU-282987 (1 µM) for 15 min; (**B**–**F**) mHypoA-29 WT and KO cells were incubated with MCM-C or MCM-LPS for 6 h. A PNU-282987 stimulus (1 µM) was administered 15 min before and 3 h after the addition of the conditioned medium; (**B**) IL-6, IL-1β (**C**), (**D**) TNF-α, (**E**) IL-10, and (**F**) Zpf36 mRNA expression was evaluated by RTq-PCR. ** *p* < 0.01, *** *p* < 0.001, n = 3–4.

**Figure 6 cells-11-02195-f006:**
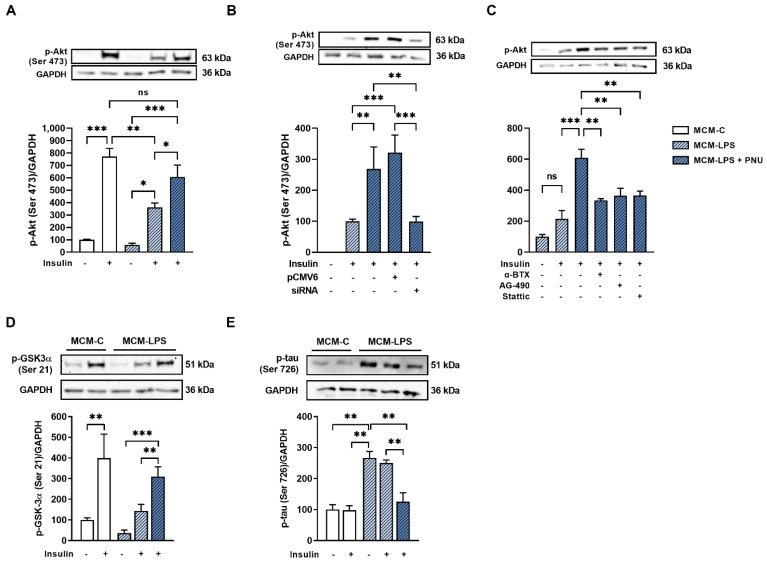
α7nAchR prevents insulin resistance and GSK-3/tau activation in neurons. mHypoA-2/29 cells were incubated for 6 h with MCM-C or MCM-LPS and treated with insulin (100 nM) for 10 min. A stimulus with PNU-282987 (1 µM) was performed 15 min before and 3 h after the addition of the conditioned medium. (**A**–**C**) Akt phosphorylation levels were assessed by Western blot (**A**); in some groups, cells were previously transfected with α7nAchR overexpression plasmid or siRNA (**B**) or treated with α-bungarotoxin (100 µM; 1h), Stattic (30 µM; 30 min), and AG-490 (100 µM; 30 min) (**C**) before the first PNU-282987 stimulus. (**D**,**E**) Western blot analysis of (**D**) GSK3 and (**E**) tau phosphorylation levels. * *p* < 0.05, ** *p* < 0.01, *** *p* < 0.001, n = 3.

**Table 1 cells-11-02195-t001:** Nutritional composition of the high-fat diet and standard chow diet.

	Standard Chow Diet ^1^	High-Fat Diet (60%)
Net protein (g%)	22.5	26.0
Fat (g%)	4.5	35.0
Carbohydrate (g%)	55.0	26.0
Crude fiber (g%)	8.0	6.0
Ash (g%)	10.0	7.0
Total	100.0	100.0
Kcal/g	3.5	5.2
Protein (kcal%) ^2^	25.7	19.9
Fat (kcal%) ^2^	11.5	60.2
Carbohydrate (kcal%) ^2^	62.8	19.9

^1^ (NUVILAB^®^ Cr-1, Nuvital, PR—Brasil). ^2^ Protein and carbohydrate = 4 kcal/g; fat = 9 kcal/g.

## Data Availability

The data presented in this study are available on request from the corresponding author.

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
