# Peer review of "Activation of the α7 Nicotinic Acetylcholine Receptor Prevents against Microglial-Induced Inflammation and Insulin Resistance in Hypothalamic Neuronal Cells"

_cells, 2022, doi:10.3390/cells11142195_

Round 1

Reviewer 1 Report

The article explores how "Activation of the α7 Nicotinic Acetylcholine Receptor Prevents Against Microglial-Induced Inflammation and Insulin Resistance in Hypothalamic Neuronal Cells". The research hypothesis is well presented. Results are clear and well discussed. However, I will leave some suggestions/comments.

Article should include in methodology, information about the gene (Chrna7) coding for the receptor (α7nAchR) and clarify through the article when they are speaking about one or the other and use uniform criteria. (for example line 120, 207, 225, 227, 229, legends and figures,...)

Wouldn't be relevant to confirm changes in the expression of the receptor by western blotting?

How was the protocol for the preparation of the conditioned medium chosen? Please provide references or previous optimization steps. 

line 229 and "mRNA" before expression

please include the rationale behind choosing p-Akt as a measure of insulin signaling instead of the ratio p-AKT/total AKT

Integration of the findings with current knowledge on the relationship between  α7 Nicotinic Acetylcholine Receptor and insulin resistance or metabolic disorders should also be discussed and references provided. 

Line 362-364 -include reference

Author Response

Response to Reviewer 1 Comments

We thank the reviewers for the critical assessment of our manuscript. In the following we address their concerns point by point

  1. Article should include in methodology, information about the gene (Chrna7) coding for the receptor (α7nAchR) and clarify through the article when they are speaking about one or the other and use uniform criteria. (for example line 120, 207, 225, 227, 229, legends and figures...)

Reply: As suggested, we have standardized the use of "Chnrna7" to refer to the gene and “α7nAchR” to refer to the protein encoded by the Chrna7 gene throughout the text and in the figure legends. A brief clarification has been added in lines 160 and 225/226.

  1. Wouldn't be relevant to confirm changes in the expression of the receptor by western blotting?

Reply: Assessment of the α7nAchR protein content by western blotting (as well as other methods based on antigen-antibody reaction, such as immunofluorescence) is not recommended because of the lack of specificity of currently commercially available α7nAchR antibodies. This explanation was added in the Discussion section (lines 407-409) and references showing the reactivity of α7nAchR in tissues from α7nAChR knockout mice (α7nAChR-/-) were added. Additionally, the assessment of mRNA levels by RT-qPCR is also not recommended for the validation of the CRISPR/Cas9 knockout, since unmodified regions of the target gene can still be transcribed normally and give rise to a truncated and non-functional protein. For this reason, we chose to confirm the loss of α7nAchR functionality in knockout cells my measuring p-STAT3 levels in response to selective agonist treatment.

  1. How was the protocol for the preparation of the conditioned medium chosen? Please provide references or previous optimization steps.

Reply: Information on the LPS treatment used and references on the adopted protocol were added (lines 124-127)

  1. Line 229 and "mRNA" before expression

Reply: The word “mRNA” was added in the text, as suggested.

  1. Please include the rationale behind choosing p-Akt as a measure of insulin signaling instead of the ratio p-AKT/total AKT

Reply: Although many authors have used the ratio p-AKT/Akt to demonstrate insulin signaling, we believed that the ratio p-AKT/endogenous control is adequate to evaluate insulin resistance. The physiological effect of AKT is performed by phosphorylated protein and the use of appropriate endogenous control (GAPDH) provides robust information on the increase of p-AKT amount in the cell after insulin stimulation.

We and other authors have used the ratio of p-AKT/endogenous control (non-phosphorylated protein/endogenous control)  in other published studies:

1-Veras ACC, Santos TD, Martins ICA, de Souza CM, Amaral CL, Franco BDS, Holanda ASS, Esteves AM, Milanski M, Torsoni AS, Ignacio-Souza LM, Torsoni MA. Low-Dose Coconut Oil Supplementation Induces Hypothalamic Inflammation, Behavioral Dysfunction, and Metabolic Damage in Healthy Mice. Mol Nutr Food Res. 2021

2- Costa SO, Souza CM, Lanza PG, Sartori JO, Ignacio-Souza LM, Candreva T, Rodrigues HG, Torsoni AS, Milanski M, Torsoni MA. Maternal high fat diet consumption reduces liver alpha7 nicotinic cholinergic receptor expression and impairs insulin signalling in the offspring. Sci Rep. 2020 Jan 8;10(1):48.

3-Sarroca et al. Resveratrol confers neuroprotection against high-fat diet in a mouse model of Alzheimer's disease via modulation of proteolytic mechanisms. J. Nutr. Biochem. 89, March 2021, 108569

4- Shen et al.,Targeted NF1 cancer therapeutics with multiple modes of action: small molecule hormone-like agents resembling the natural anticancer metabolite, 2-methoxyoestradiolBr J Cancer. 2015 Oct 20;113(8):1158-67.

5- Aboulkassim et al.. Ligand-dependent TrkA activity in brain differentially affects spatial learning and long-term memory. Mol Pharmacol. 2011 Sep;80(3):498-508

  1. Integration of the findings with current knowledge on the relationship between α7 Nicotinic Acetylcholine Receptor and insulin resistance or metabolic disorders should also be discussed and references provided.

Reply: An integration between our findings and recently published works has been added in the Discussion section (lines 441-448)

  1. Line 362-364 -include reference

Reply: References were included, as suggested.

Reviewer 2 Report

This study by Libardi do Amaral et al. aimed to study the role of a7nAChR in neuronal cells in the development of insulin resistance in an inflammatory context. Using in vitro models showed that exposure to microglia conditioned medium induced the expression of pro-inflammatory markers in hypothalamic neurons. The data suggest that the activation of a7nAChR can blunt this response by inhibiting NFKB translocation to the nucleus and the increased gene expression of IL10 and TTP, ultimately improving insulin resistance in neurons. The topic is of interest and although several of the initial findings are somewhat incremental others are novel and could make a solid contribution to the literature. Several major concerns are listed below: 

- The blots for the non-phosphorylated protein levels should be included in the figures. 

- In the Material and Method section the authors stated that experiments were performed 3 times independently, but some figures showed an N of 4 (I.e.: F3d, f). 

- Figure 3. IL6 expression in mHypoA-2/29 is similar when control cells are treated by MCM-LPS or LPS. However, IL1b and Tnf expression are markedly increased by LPS treatment in comparison with MCM-LPS. Have the authors some insights about it?  

-In line 289, the authors stated that “Treatment with PNU prevented MCM-LPS-induced 289 increased gene expression of IL-6, IL-1β, TNF-α, and CX3CL1 (Figure 4 A-D)”, but IL6 and TNF, even reduced in comparison with MCM-LPS group, still showed increased expression levels when compared with the control group. This statement should be toned down. 

- Why the knockout confirmation of a7nAChR in mHypoA-2/29 was done indirectly by analyzing pSTAT3 protein levels? Please show the knockout of a7nAChR. 

- The MCM-C control group is missing in the Figure 5F panel. 

Minor 

-Line 143, did the authors mean AG-490? 

-In figure 6D, the statistic between MCM-C groups is missing. 

Author Response

Response to Reviewer 2 Comments

We thank the reviewers for the critical assessment of our manuscript. In the following we address their concerns point by point

  1. The blots for the non-phosphorylated protein levels should be included in the figures.

Reply: Although many authors have used the ratio phosphorylated protein/non-phosphorylated protein, the ratio of non-phosphorylated protein/endogenous control is adequate to assess the pathway's ability to phosphorylate a given protein. Besides the use of appropriate endogenous control (GAPDH or actin) provides robust information of phosphorylated protein level in the cell after challenge.

We and other authors have used the ratio non-phosphorylated protein/endogenous control in other published studies:

1-Veras ACC, Santos TD, Martins ICA, de Souza CM, Amaral CL, Franco BDS, Holanda ASS, Esteves AM, Milanski M, Torsoni AS, Ignacio-Souza LM, Torsoni MA. Low-Dose Coconut Oil Supplementation Induces Hypothalamic Inflammation, Behavioral Dysfunction, and Metabolic Damage in Healthy Mice. Mol Nutr Food Res. 2021

2- Costa SO, Souza CM, Lanza PG, Sartori JO, Ignacio-Souza LM, Candreva T, Rodrigues HG, Torsoni AS, Milanski M, Torsoni MA. Maternal high fat diet consumption reduces liver alpha7 nicotinic cholinergic receptor expression and impairs insulin signalling in the offspring. Sci Rep. 2020 Jan 8;10(1):48.

3-Sarroca et al. Resveratrol confers neuroprotection against high-fat diet in a mouse model of Alzheimer's disease via modulation of proteolytic mechanisms. J. Nutr. Biochem. 89, March 2021, 108569

4- Shen et al.,Targeted NF1 cancer therapeutics with multiple modes of action: small molecule hormone-like agents resembling the natural anticancer metabolite, 2-methoxyoestradiol. Br J Cancer. 2015 Oct 20;113(8):1158-67.

5- Aboulkassim et al.. Ligand-dependent TrkA activity in brain differentially affects spatial learning and long-term memory. Mol Pharmacol. 2011 Sep;80(3):498-508

  1. In the Material and Method section the authors stated that experiments were performed 3 times independently, but some figures showed an N of 4 (I.e.: F3d, f).

Reply: The sentence was changed in line 212 to "at least three independent experiments".

  1. Figure 3. IL6 expression in mHypoA-2/29 is similar when control cells are treated by MCM-LPS or LPS. However, IL1b and Tnf expression are markedly increased by LPS treatment in comparison with MCM-LPS. Have the authors some insights about it?

Reply: Thanks for the careful review and the important question about the manuscript. The IL6 expression shown in figure 3 was evaluated in two experiments (fig. 3A and 3G). In Figure 3A it is possible to observe that exposure to MCM-LPS significantly increases the amount of mRNA for IL6. However, in fig3G, as pointed out by the reviewer, LPS did not induce further increase. Initially, this protocol (fig.3G-3I) aimed to assess whether there was a residual presence of LPS in the medium from BV cells (MCM-LPS). However, this experiment also suggests that IL6 expression in neuronal cells (mHypoA-2/29), induced by molecules secreted by BV cells after LPS treatment, does not depend on the TLR4 pathway. As shown by the lack of effect of the TLR4 inhibitor (TAK-242) when administered prior to MCM-LPS treatment. It is possible that cytokines and chemokines secreted by BV cells and present in MCM-LPS were able to robustly induce IL6 expression in mHypoA-2/29. Thus, treatment with LPS directly on mHypoA-2/29 cells did not result in additional IL6 expression, suggesting that IL1b and Tnf are more responsive to LPS/TLR4 pathway than IL6.

  1. In line 289, the authors stated that “Treatment with PNU prevented MCM-LPS-induced increased gene expression of IL-6, IL-1β, TNF-α, and CX3CL1 (Figure 4 A-D)”, but IL6 and TNF, even reduced in comparison with MCM-LPS group, still showed increased expression levels when compared with the control group. This statement should be toned down.

Reply: The sentence was corrected in line 289 to "Treatment with PNU attenuated MCM-LPS (…)".

  1. Why the knockout confirmation of a7nAChR in mHypoA-2/29 was done indirectly by analyzing pSTAT3 protein levels? Please show the knockout of a7nAChR.

Reply: As mentioned above in response to Revisor 1, the knockout confirmation was done indirectly due to the unreliability of commercially available α7nAchR antibodies.

  1. The MCM-C control group is missing in the Figure 5F panel.

Reply: Thanks for the careful review. The group MCM-C was not included in the Fig.5F because in the previous experiment (fig.4F) the expression of Zfp36 in MCM-C and MCM-LPS was not different.

  1. Line 143, did the authors mean AG-490?

Reply: Yes, the spelling mistake was corrected.

  1. In figure 6D, the statistic between MCM-C groups is missing.

Reply: We added the statistics between MCM-C groups.

Round 2

Reviewer 2 Report

All questions have been addressed.